

# Resequencing and characterization of the first *Corynebacterium pseudotuberculosis* genome isolated from camel

Enrico Giovanelli Tacconi Gimenez[1,*], Marcus Vinicius Canário Viana[1,*], Thiago de Jesus Sousa[2], Flávia Aburjaile[3], Bertram Brenig[4], Artur Silva[5] and Vasco Azevedo[1]

[1] Institute of Biological Sciences, Federal University of Minas Gerais, Belo Horizonte, Minas Gerais, Brazil
[2] Laboratório Central do Espírito Santo (LACEN-ES), Vitória, Espírito Santo, Brazil
[3] Veterinary School, Federal University of Minas Gerais, Belo Horizonte, Minas Gerais, Brazil
[4] Institute of Veterinary Medicine, University of Göttingen, Göttingen, Niedersachsen, Germany
[5] Institute of Biological Sciences, Federal University of Pará, Belém, Pará, Brazil
[*] These authors contributed equally to this work.

Corresponding author
Vasco Azevedo,
vascoariston@gmail.com

## ABSTRACT

**Background.** *Corynebacterium pseudotuberculosis* is a zoonotic Gram-positive bacterial pathogen known to cause different diseases in many mammals, including lymph node abscesses in camels. Strains from biovars equi and ovis of *C. pseudotuberculosis* can infect camels. Comparative genomics could help to identify features related to host adaptation, and currently strain Cp162 from biovar equi is the only one from camel with a sequenced genome.

**Methods.** In this work, we compared the quality of three genome assemblies of strain Cp162 that used data from the DNA sequencing platforms SOLiD v3 Plus, IonTorrent PGM, and Illumina HiSeq 2500 with an optical map and investigate the unique features of this strain. For this purpose, we applied comparative genomic analysis on the different Cp162 genome assembly versions and included other 129 genomes from the same species.

**Results.** Since the first version of the genome, there was an increase of 88 Kbp and 121 protein-coding sequences, a decrease of pseudogenes from 139 to 53, and two inversions and one rearrangement corrected. We identified 30 virulence genes, none associated to the camel host, and the genes *rpob2* and *rbpA* predicted to confer resistance to rifampin. In comparison to 129 genomes of the same species, strain Cp162 has four genes exclusively present, two of them code transposases and two truncated proteins, and the three exclusively absent genes *lysG*, NUDIX domain protein, and Hypothetical protein. All 130 genomes had the rifampin resistance genes *rpob2* and *rbpA*. Our results found no unique gene that could be associated with tropism to camel host, and further studies should include more genomes and genome-wide association studies testing for genes and SNPs.

# INTRODUCTION

*Corynebacterium pseudotuberculosis* is a zoonotic Gram-positive bacterium that causes caseous lymphadenitis (CLA) in various animals, including small ruminants, cattle, camelids, and other host disease manifestations. In this species, biovar equi is nitrate positive and biovar ovis is nitrate negative (*Dorella et al., 2006*). In Australian sheep, CLA causes estimated losses of $A12–$A15 million and $A17 million per year for the meat and wool industry, respectively (*Baird & Fontaine, 2007*). In Australia, the wild dromedary population in the interior has frequently exhibited unsightly lymph node abscesses. Similarly, in East Africa, a high prevalence of swollen external lymph nodes has been observed in almost all dromedaries, and it is believed that this may be linked to CLA resulting from the consumption of thorny plants. CLA mortality rates in dromedaries in countries other than Europe, where it can reach 15%, are unknown. However, death always occurs when the pathogen spreads into internal organs, mainly the lung and liver (*Wernery & Kinne, 2016*).

In this context, genomic data can be used for identification and taxonomy (*Parks et al., 2022*), evolutionary studies (*Sheppard, Guttman & Fitzgerald, 2018*), epidemiology (*Gardy & Loman, 2018*), and the development of control mechanisms such as drugs (*Serral et al., 2021*) and vaccines (*Goodswen, Kennedy & Ellis, 2023*). An ideal genome assembly should be complete, closed, and artifacts-free to avoid bias in analysis that relies on gene content, variant calling (*Di Marco et al., 2023*) and synteny (*Mascher & Stein, 2014*; *Yuan, Chung & Chan, 2020*). The highly accurate but short reads from recent second-generation DNA sequencing platforms result in assembly gaps with long repetitive sequences (*Loman et al., 2012*). Two strategies are used to solve this issue using short-read data: scaffolding using an optical map (*Lehri, Seddon & Karlyshev, 2017*) and a hybrid assembly, in which longer reads with lower accuracy from a third-generation DNA sequencing platform are used to generate an assembly that is error corrected using reads from a second-generation platform (*Craddock et al., 2022*; *Di Marco et al., 2023*). With the increasing quality of long-read sequencing and assembly algorithms, long-reads alone can be used for genome sequencing (*Nurk et al., 2022*; *Sereika et al., 2022*).

In *C. pseudotuberculosis,* it is known that horses and buffalo are only reported as hosts of the nitrate-positive biovar equi, but little is known about the mechanisms related to host tropism, besides the suggestion of diphtheria toxin as a requirement to infect buffalo (*Viana et al., 2017*). Strain Cp162 from the camel is currently the only strain from the camel with a sequenced genome. It was initially isolated from an external neck abscess of a camel in 1999 and was first sequenced in 2012 using the platform SOLiD v3 Plus (RefSeq accession NC_018019.1) (*Hassan et al., 2012*). The genome was then resequenced in 2017 using the Ion Torrent PGM platform (NC_018019.2) to improve genome assembly quality, and in 2019 using Illumina HiSeq 2500 with assembly using an optical map to improve the accuracy of the genome assembly (NC_018019.3) (*Sousa et al., 2019*).

In this study, we aimed to evaluate the improvements in the genome assemblies of strains Cp162, search for genes that could be related to tropism for camels and update the pangenome analysis of the species.

## MATERIALS & METHODS

### Samples, quality assessment, and taxonomy

The genome sequences of 142 *C. pseudotuberculosis* strains were obtained from the NCBI RefSeq Database (https://www.ncbi.nlm.nih.gov/datasets/genome/?taxon=1719). From those, we removed 10 mutant strains (SigH, SigmaE, SigM, sigC, T1, Cp13, sigB, SigD, phoP, SigK) and strains 1002 and DSM 20689 due to those being the same strains as 1002B and ATCC19410, respectively. A total of 130 strains remained for downstream analysis (Data S1). The genome assemblies in fasta, gbff and gff format were retrieved using NCBI Datasets v15.6.1 (https://github.com/ncbi/datasets) with an input file containing the list of RefSeq assembly accession numbers. The current assembly of Cp162 is on version 3. The first and second assemblies of Cp162 (GCF_000265545.1 and GCF_000265545.2) were added for completeness of the dataset, but they were not included in the species analysis due to potential sequencing errors and misassemblies. CheckM2 v1.0.2 (https://github.com/chklovski/CheckM2) was used to evaluate the completeness and contamination of the genome sequences, while GUNC v1.0.5 (*Orakov et al., 2021*) was used to identify chimeric contigs, both with standard parameters. Taxonomic classification was performed using GTDB-Tk v2.3.0 with database R214 (*Chaumeil et al., 2022*) with the "Classify workflow" (classify_wf) and skipping ANI screen (–skip_ani_screen).

### Analysis of Cp162 genome assemblies

We compared the three versions of Cp162 assemblies for completeness and contamination, size, gene content, and synteny. The number of genes was collected from the GBFF file of each genome. Differences in gene content were identified using Panaroo v1.3.3 (*Tonkin-Hill et al., 2020*) for gene clustering, with the parameters "–remove-invalid-genes" and "–clean-mode strict". An in-house script was used for identifying exclusive genes (Data S2). Synteny was verified using Mauve v20150226 (*Darling et al., 2004*) with progressiveMauve algorithm.

Genome characterization was performed on the latest genome assembly (GCF_000265545.3). For mobile elements, prophages were predicted using the online tool PHASTER (https://phaster.ca/) (*Arndt et al., 2016*), while Genomic Islands (GEIs) were predicted using GIPSy v1.1.3 (*Soares et al., 2016*) and *C. glutamicum* (NZ_CP025533.1) as a reference genome. Virulence genes were predicted using PanViTa v1.1.3 (*Rodrigues et al., 2023*) with the VFDB database (-vfdb) (*Liu et al., 2022*). As PanViTa uses the core dataset of VFDB, we modified it to use the full dataset (Data S3) Antimicrobial resistance genes were predicted using PanViTa with the CARD database (-card) (*Alcock et al., 2023*). CRISPR-Cas systems were predicted using the online tool CRISPRCasFinder v1.1.2 (https://crisprcas.i2bc.paris-saclay.fr/CrisprCasFinder/Index) (*Couvin et al., 2018*) with standard parameters. A circular map comparing the three assemblies of Cp162 was built using BRIG v0.95 (*Alikhan et al., 2011*). The GBFF file was used as input for PHASTER, GIPSy and PanViTa, while the nucleotide fasta file was used as input for Mauve and BRIG.

## Species-level analysis

To build a phylogenomic tree, we used Panaroo to identify the shared protein-coding genes across the 130 *C. pseudotuberculosis* isolates, and the outgroup *C. ulcerans* NCTC 7910 (GCF_900187135.1) and perform a multiple sequence alignment (MSA) using MAFFT (*Katoh et al., 2005*). The parameters used were "–remove-invalid-genes–clean-mode strict -a core–core_threshold 0.95–aligner mafft". The phylogenetic inference was built from the MSA using IQ-TREE2 v2.0.7 (*Minh et al., 2020*) with standard parameters, which includes the maximum-likelihood (ML) method in which the best-fit model of nucleotide substitution selected automatically by ModelFinder (*Kalyaanamoorthy et al., 2017*). Support values were calculated using ultrafast bootstrap approximation with 1,000 replicates (-B 1000) (*Hoang et al., 2018*). The tree was visualized and annotated using the online tool Interactive Tree of Life (iTOL) v6.8 (https://itol.embl.de/).

A pangenome is a set of non-redundant genes that composes the repertoire of all genomes of a species (*Tettelin et al., 2005*). The pangenome and distribution of genes across all the 130 strains were identified using Panaroo due to its feature of "refinding" genes that were not annotated due to annotation artifacts. In this software, the genes are classified by frequency in the categories core genes (99% ≤ strains ≤ 100%), softcore genes (95% ≤ strains < 99%), shell genes (15% ≤ strains < 95%), cloud genes (0% ≤ strains < 15%) and total genes (0% ≤ strains ≤ 100%) (*Tonkin-Hill et al., 2020*). The pangenome development was calculated using Heap's Law formula, implemented in the R package Micropan v2.1 (https://github.com/larssnip/micropan) to estimate whether it is an open or closed pangenome using 10,000 permutations (nper = 10,000). The gene clusters were annotated using eggNOG-mapper v2.1.9 with database v5.0.2 (*Huerta-Cepas et al., 2017*) with standard parameters. The presence of virulence and antimicrobial resistance genes were verified using PanViTa with VFDB and CARD databases.

# RESULTS

## Quality assessment and taxonomy

All genomes were classified as *C. pseudotuberculosis*. Completeness and contamination ranged between 97.37% and 100% and 0.14% and 6.43%, respectively. No evidence of chimerism was detected by GUNC (Data S1).

## Analysis of Cp162 genome assemblies

The comparisons between the three assembly versions showed increased genome size and the number of coding sequences (CDS) (Table 1). Across all three versions, we identified 2,128 genes, 2,010 of them shared. About virulence genes, the first assembly version lacks the virulence genes *nanH* while *spaC* is fragmented as CP162_RS09080 and CP162_RS09085). The synteny analysis showed two inversions and one rearrangement in the first version, which were later corrected in the subsequent versions using the optical map (Fig. 1). Since the first assembly, the genome had an increase of 88 Kbp and 121 CDSs and a decrease of pseudogenes from 139 to 53. The third version has 18 exclusively present genes (Table 1, Data S4).

**Table 1 Comparison between the three versions of the strain Cp162 genome assembly.**

|  | Version 1 | Version 2 | Version 3 |
|---|---|---|---|
| Deposit date | 01/31/2014 | 07/15/2017 | 12/16/2019 |
| Platform | SOLiD | IonTorrent | Illumina HiSeq 2500 |
| Coverage | 686x | 200x | 713x |
| Size (bp) | 2,293,464 | 2,365,874(+72,410) | 2,382,183(+88,639) |
| Completeness (%) | 99.9 | 99.9 | 99.9 |
| Contamination (%) | 0.21 | 0.2 | 0.23 |
| CDSs | 2,043 | 2,112 (+69) | 2,164 (+121) |
| Exclusively present CDSs | 3 | 2 | 18 |
| Exclusively absent CDSs | 88 | 1 | 6 |
| Pseudo Genes (total) | 139 | 80 | 53 |
| Pseudo Genes (ambiguous residues) | 0 | 0 | 0 |
| Pseudo Genes (frameshifted) | 116 | 67 | 41 |
| Pseudo Genes (incomplete) | 13 | 9 | 7 |
| Pseudo Genes (internal stop) | 17 | 9 | 6 |
| Pseudo Genes (multiple problems) | 7 | 5 | 1 |
| tRNA | 49 | 49 | 63 |
| 5S rRNA | 4 | 4 | 4 |
| 16S rRNA | 4 | 4 | 4 |
| 23S rRNA | 4 | 4 | 4 |
| ncRNA | 3 | 3 | 3 |
| Virulence genes | 31 ($srtB^+$, $sapA^-$) | 31 ($srtB^-$, $sapA^+$) | 30 ($srtB^-$, $sapA^-$) |
| Antimicrobial resistance genes | $rpoB2$, $rbpA$ | $rpoB2$, $rbpA$ | $rpoB2$, $rbpA$ |

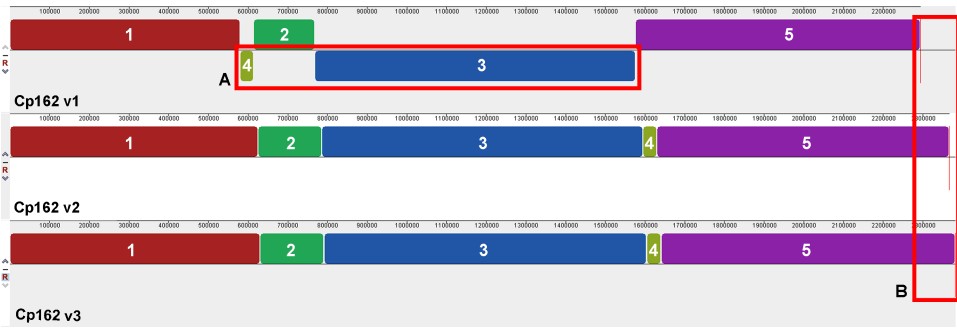

**Figure 1 Alignment of the three genome assembly versions of *Corynebacterium pseudotuberculosis* strain Cp162.** (A) Two inversions and one rearrangement in the first assembly. (B) Increase in genome size throughout the assemblies.

In the third assembly version, we predicted one incomplete prophage with 8.9 Kb and 18 CDSs (Data S5, S6 and S7), 13 GEIs, five of them pathogenicity islands (Fig. 2, Data S7), three CRISPR arrays, and a Type I-E CRISPR-Cas system (Data S7). PanViTa identified 30 virulence genes. The first and second assembly versions had *strB* and *sapA* as exclusive virulence genes, respectively (Table 1, Data S8). The analysis of Mauve alignment

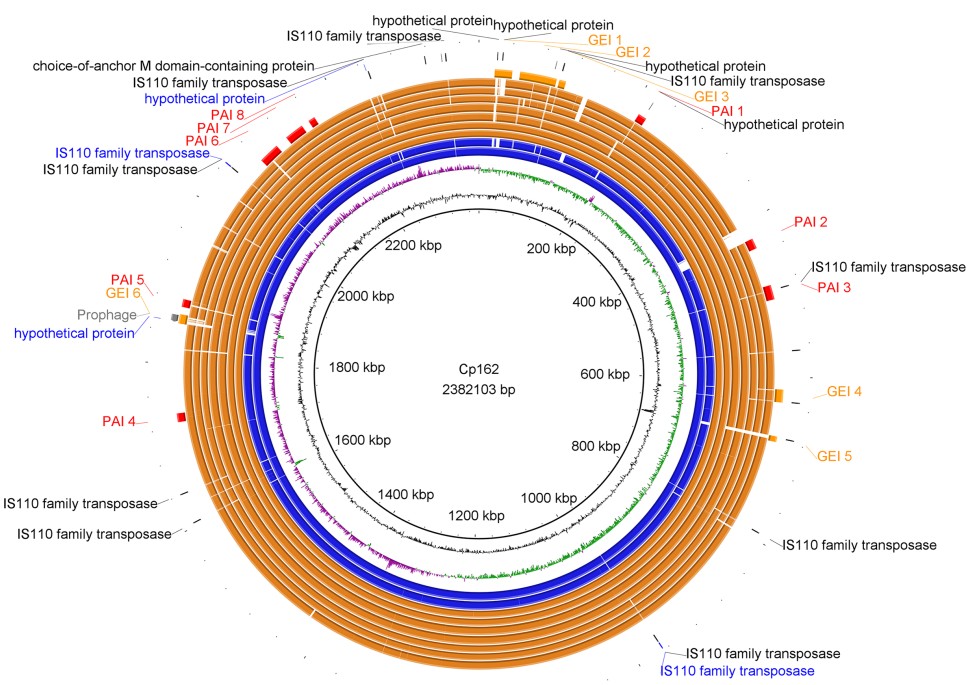

**Figure 2** **Circular map of *Corynebacterium pseudotuberculosis* Cp162 (camel).** From inner to outer circle: Cp162 v3 (equi, camel), CG Content, GC Skew, Cp162 v2, Cp162 v1, I31 (equi, cow), G1 (equi, alpaca), 31 (equi, buffalo), 258 (equi, cow), 262 (equi, cow), I19 (ovis, cow), 1002B (ovis, goat), genomic islands (GEI) and pathogenicity islands (PAI), prophage, and exclusive genes of Cp162 v3 in comparison to v2 and v1, and exclusive genes of Cp162 v3.

revealed that the CDSs classified as *srtB* (v1 locus_tag: CP162_RS09100, old_locus_tag: Cp162_1849) and *sapA* (v2 locus_tag: CP162_RS09670, old_locus_tag: Cp162_1968) were present in the three assembly versions, but with variations in size. PanViTa identified two antimicrobial resistance proteins: Rifampin-resistant beta-subunit of RNA polymerase (*rpoB2*, WP_041481489.1) and RbpA bacterial RNA polymerase-binding protein (*rpbA*, WP_014800420.1) (Table 1, Data S8).

## Phylogeny and pangenome

A tree was generated by IQ-TREE2 using core genome alignment from Panaroo and MAFFT. The species tree in Fig. 3 shows two main clades. The first one contains a subclade composed of strains Cp162 (camel), G1 (alpaca), and I37 (cow, Israel) and another subclade containing strains isolated from horses and buffalo, separated by a host. The second one contains strain 262 (cow, Belgium) and all biovar ovis strains (collapsed) as its sister group. In the pangenome analysis, 2,332 genes were identified in the pangenome ($\alpha = 1.27$), 1,877 as core, 68 as softcore, 173 as shell, and 214 as cloud. Of 2,332 genes, 2,181 were scanned by eggNOG-mapper, and 1,953 had a functional annotation (Data S9). The PanViTa analysis showed the presence of the virulence genes *tufA*, DIP_RS14950, *mprA* in most genomes and the diphtheria toxin gene (*tox*) only in genomes from strains isolated from buffalo

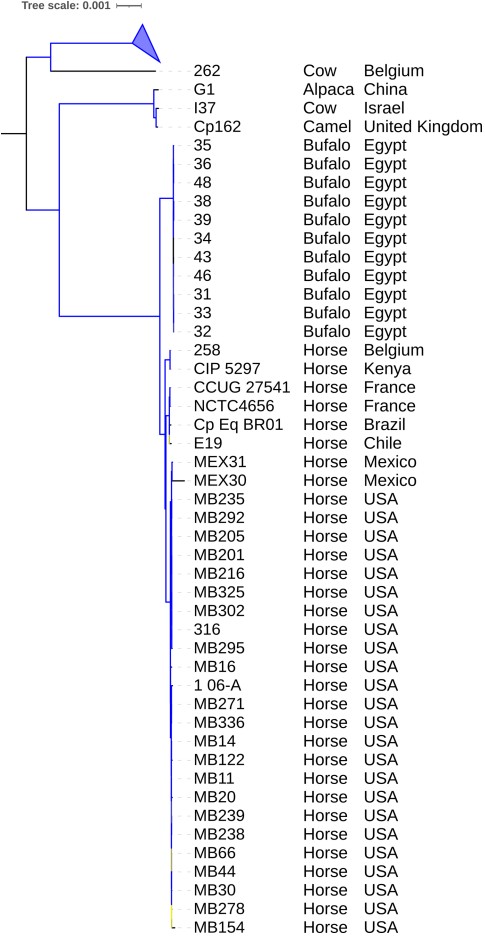

**Figure 3** **Phylogenomic tree of *Corynebacterium pseudotuberculosis* genomes.** The tree was built using the core genome identified and aligned using Panaroo and MAFFT, respectively. A phylogeny using the Maximum Likelihood method was built using IQ-TREE2 with 1,000 replicates of ultrafast bootstrap approximation and *C. ulcerans* NCTC7910 (not shown) as an outgroup. Bootstrap values are represented as a branch color scale that ranges from 85% (yellow) to 100% (blue). The biovar ovis clade is collapsed.

(strains 31, 32, 33, 34, 35, 36, 38, 39 and 48). The antimicrobial resistance genes *rpoB2* and *rbpA* were present in all genomes (Data S8).

## Exclusive genes of Cp162

From the pangenome analysis, we also identified four proteins exclusively present in Cp162 and three exclusively absent in this lineage (Table 2). In the exclusively present group, two were predicted as the same transposase (WP_048653436.1). The other two are truncated (41 and 45 aa), none showed conserved domains, and one is in GEI 6. Analysis against the GenBank database using BLASTp and WP_275060758.1 as a query showed 92% of coverage with 52% of identity to an ATP-dependent helicase of *Streptomyces spp.* The same analysis with WP_231131458.1 showed 97.8% of identity with other truncated proteins from CpE19_1664 (AKS14002.1).

**Table 2  Exclusively present and absent genes in *Corynebacterium pseudotuberculosis* strain Cp162 (camel) in comparison to other 129 genomes of the same species.**

| Locus Tag (protein ID) | Gene | Product | GEI | Functional annotation |
|---|---|---|---|---|
| Exclusively present | | | | |
| CP162_RS04525 (WP_048653436.1) | tnp3510a | IS110 family transposase | – | COG: L, Pfam: DEDD_Tnp_IS110, Transposase_20 |
| CP162_RS09445 (WP_048653436.1) | tnp3510a | IS110 family transposase | – | – |
| P162_RS11030 (WP_231131458.1) | – | Hypothetical protein | GEI 6 | – |
| CP162_RS11150 (WP_275060758.1) | – | Hypothetical protein | – | – |
| | | | | |
| Exclusively absent | | | | |
| (WP_014366903.1) | *lysG* | Transcriptional regulator | – | COG: K, KEGG: K05596, PFAM: HTH_1, LysR_substrate |
| (AKC74244.1) | – | NTP pyrophosphohydrolases, including oxidative damage repair enzymes | – | COG: L, PFAM: NUDIX |
| (WP_038617038.1) | – | – | – | – |

**Notes.**
COG, Cluster of Orthologous Genes; GEI, Genomic Island; KEGG, Kyoto Encyclopedia of Genes and Genomes.

From the group of proteins exclusively absent in Cp162, one was recognized by eggNOG-mapper as a Transcriptional Regulator named *lysG* (COG category: K), another as an enzyme from NUDIX superfamily (COG category: L), and the last one as a hypothetical protein with no domains.

# DISCUSSION

The improvements in long-read DNA sequencing platforms allowed the assembly of complete or near complete bacterial genomes (*Sereika et al., 2022*) and may eliminate in the future the use of optical map for contig ordering and short-reads for lower error rates. Currently, the combination of long- and short-reads is still relevant (*Di Marco et al., 2023*; *Hepner et al., 2023*).

Our results showed that using Illumina HiSeq and an optical map increased the genome size and number of CDSs, corrected misassembles, and reduced the number of pseudogenes (Table 1). Concerning synteny, the correct sequence order and content are required to study the genome plasticity events such as inversion, translocation, insertion, and deletions (*Lehri, Seddon & Karlyshev, 2017*). As shown in Fig. 1, optical mapping could correctly order contigs from sequencing. The rearranged regions are strictly between transposase genes, which could explain possible rearrangements (*Hickman & Dyda, 2016*). Some transposase sequences were found only in the third version of the Cp162 genome, within its exclusive 18 genes (Data S4). With frameshifts, the first and second genome versions were sequenced using SOLiD and Ion Torrent platforms, known for indel sequencing errors (*Loman et al., 2012*), leading to CDS frameshifts. The correct identification of pseudogenes is required for gene evolution analysis and for gene content studies such as pangenomics. In NCBI's PGAP annotation pipeline (*Li et al., 2021*), a pseudogene will not have a CDS annotation, while in the RAST-Tk pipeline (*Brettin et al., 2015*), each fragment of a pseudogene can be

annotated as a CDS; this can lead to erroneous estimations of gene content across genomes and suggests that data generated using SOLiD and Ion Torrent should be used with caution.

In relation to virulence factors, we identified 30 in Cp162, and 35 distributed across the other genomes, with *tox* exclusively in strains isolated from buffalo (Table 1, Data S8). We could not associate a virulence gene to the camel isolate Cp162. In buffalo, the presence of the *tox* was suggested as a requirement to infect this host (*Viana et al., 2017*).

In relation to antimicrobial resistance genes, Cp162 and all the other genomes have the genes *rpoB2* and *rbpA* (Data S8) that confer resistance to rifampin according to the CARD database (CARD accessions ARO:3000501 and ARO:3000245). Although rifampin mixed with other types of antimicrobials have been suggested by some authors (*Heggelund et al., 2015*; *Sting et al., 2022*), a recent study pointed that resistance to rifampin is present in some lineages infecting sheep and goats (*El Damaty et al., 2023*). This result suggests this antimicrobial should be avoided for infection treatment. Although *C. pseudotuberculosis* is susceptible to many antibiotic chemicals *in vitro*, the intracellular nature and encapsulation around lesions confers some protection (*Baird & Fontaine, 2007*). Common antibiotics used for treatment are penicillin or erythromycin combined with rifampin (*Williamson, 2001*). In the case of particularly valuable animals, surgical treatment of the lesions can be performed (*Baird & Fontaine, 2007*).

In Cp162 we predicted an incomplete prophage (Data S5 and S6) and 13 GEIs (Fig. 2, Data S7). GEI 5 is exclusive to the clade composed of Cp162 (camel), I37 (cow), and G1 (alpaca) (Fig. 2), but it may be due to a common ancestor rather than host tropism because strain 262 (equi) and I19 (ovis) also infect cows. The genome has three CRISPR arrays and a Type I-E CRISPR-Cas system (Table 1). Type I-E was previously found only in biovar equi in *C. pseudotuberculosis*, while proteins from Type III restriction-modification systems were exclusive from biovar ovis (*Parise et al., 2018*).

Cp162 is the only strain from a camel with a sequenced genome, and we looked for genes that could be involved in the tropism of this host by comparing its genome to 129 others from the same species. The exclusively present genes are transposases and truncated proteins, while the exclusively absent are *lysG*, an enzyme from the NUDIX superfamily and a hypothetical protein with no domains (Table 2). There is no clear relation between those genes and host tropism for camels. If there are any genome features related to tropism, they could be verified by sequencing the genomes of more strains from this host and performing a genome-wide association study (GWAS) testing for gene presence/absence or SNPs.

The phylogeny of 130 genomes (Fig. 3) supports the previous assumption that biovar ovis is a clade that originated from biovar equi, with its exclusive adaptations, and biovar equi as paraphyletic with two exclusive hosts (horse and buffalo) (*Viana et al., 2018*). Sampling more strains from camels could show they form exclusive clades in biovar equi and ovis that could suggest clonal expansion after host adaptation. The species pan-genome was estimated as closed ($\alpha > 1.00$), which means that sequencing more genomes will not reveal new genes (*Tettelin et al., 2008*).

## CONCLUSIONS

The genome resequencing of strain Cp162 and assembly using an optical map resulted in corrections of synteny and fewer pseudogenes caused by sequencing artifacts. The comparative analysis suggests that there are no genes related to the tropism for camels, but this could be tested again using more genomes from this host and performing association tests for genes and nucleotide variations.

### Funding

This work was supported by the Federal University of Minas Gerais, Fundação de Amparo à Pesquisa do Estado de Minas Gerais (FAPEMIG) and Federal University of Pará. The funders had no role in study design, data collection and analysis, decision to publish, or preparation of the manuscript.

### Grant Disclosures

The following grant information was disclosed by the authors:
The Federal University of Minas Gerais, Fundação de Amparo à Pesquisa do Estado de Minas Gerais (FAPEMIG) and Federal University of Pará.

### Competing Interests

Vasco Azevedo is an Academic Editor for PeerJ.

### Author Contributions

- Enrico Giovanelli Tacconi Gimenez performed the experiments, analyzed the data, prepared figures and/or tables, authored or reviewed drafts of the article, and approved the final draft.
- Marcus Vinicius Canário Viana performed the experiments, analyzed the data, prepared figures and/or tables, authored or reviewed drafts of the article, and approved the final draft.
- Thiago de Jesus Sousa conceived and designed the experiments, performed the experiments, analyzed the data, authored or reviewed drafts of the article, and approved the final draft.
- Flávia Aburjaile analyzed the data, authored or reviewed drafts of the article, and approved the final draft.
- Bertram Brenig performed the experiments, authored or reviewed drafts of the article, and approved the final draft.
- Artur Silva conceived and designed the experiments, performed the experiments, analyzed the data, authored or reviewed drafts of the article, and approved the final draft.
- Vasco Azevedo conceived and designed the experiments, performed the experiments, analyzed the data, authored or reviewed drafts of the article, and approved the final draft.
## Data Availability

We used 142 genomes available in GenBank (Table S1).

## Supplemental Information

Supplemental information for this article can be found online at http://dx.doi.org/10.7717/peerj.16513#supplemental-information.

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
