# Peer review of "Resequencing and characterization of the first Corynebacterium pseudotuberculosis genome isolated from camel"

_PeerJ, doi:10.7717/peerj.16513_

## Round 0.1 · original submission · Major Revisions

Three referees have evaluated the manuscript. One critical concern is to provide more detail on the methodology, including data and analysis pipeline sharings, along with modification of the text according to suggestions noted. Providing more discussion to address reviewers' concerns is also advised.

·

Basic reporting

The paper is well written and easy to follow. The authors clearly communicate their goal of comparing 3 existing assemblies to demonstrate that combining optical mapping with short read assembly can improve assembly quality and gene retrieval, thus aiding in comparative genomics. The manuscript builds on previous work of the authors in which they perform comparative analysis across several C. pseudotuberculosis assemblies. The authors report no significant novel insights from the current best assembly, i.e. no new genes in Cp162 strain related to tropism in camels, and propose further investigation by sequencing more genomes from this host & performing GWAS analysis. The figures & tables are clear (although there could be some minor improvements made as suggested further below), and data & software settings are provided for reproducibility.

Suggested Improvement:
figure 1: Please clearly label the 3 assemblies, especially considering they are in reverse order, i.e. Cp162_v3.fasta is presented first. For comparison, a similar figure previously generated by authors for Cp162 (Figure 6 in Sousa T de J 2019) is much clearer.

figure 2 & figure 3: The caption for figure 2 in the manuscript appears to be swapped with the caption for figure 3

In table 1: IonTorrent is misspelled

Experimental design

The authors have taken a well studied approach to comparative genomics and their methods used are reasonable. They make good use of previously generated data from the NCBI RefSeq Database, and adopt standard tools in their pipeline for checking errors, taxonomic classification, synteny detection, pangenomics and other steps.

Validity of the findings

The results are clear and findings are valid. The primary observation of the authors -- improved assembly when combining short read assembly with optical mapping, is in-keeping with earlier findings. Phylogenetic analysis is consistent with earlier work as well. Comparative analysis with other strains of C. pseudotuberculosis did not find exclusively present genes in Cp162 that might be associated with host tropism for camels.

Additional comments

The authors propose two strategies to produce complete genome assemblies: 1. scaffolding using an optical map, and 2. hybrid assembly using a mix of high quality short reads & error prone long reads. In addition, there is currently a widely accepted 3rd strategy which is to purely use high quality long reads at low coverage, which has been shown to generate complete assemblies (Nurk S, et al. The complete sequence of a human genome. Science. 2022 Apr;376(6588):44-53). Going forward, would such an approach remove the need for optical mapping based correction of assemblies? Please comment.

Reviewer 2 ·

Basic reporting

Review for Gimenez et al “Resequencing of Corynebacterium pseudotuberculosis Cp162 genome and the search for host tropism mechanisms”.

Gimenez et al Have worked on Corynebacterium pseudotuberculosis strain Cp162 and compared it with the three available genome assemblies. The strain Cp162 is the only sequenced genome found in camel. The authors describe the changes in the genome in detail since the sequencing of the first genome. However, they did not find any gene responsible for host tropism when compared with other available 129 genomes.
The paper is short crisp and needs minor improvements.



Minor comments

Consider changing the title as it includes the word mechanism, but no mechanism of host tropism was explained by this study in the paper.

Line 55 Use consistent font size for citations (Wernery & Kinne, 2016) Same for
Line 57 citation (Sheppard, Guttman & Fitzgerald, 2018) and rest of the manuscript.
Line 61 Please add a reference for “synteny”….
“An ideal genome assembly should be complete, closed, and artifacts-free to avoid bias in analysis that relies on gene content, variant calling (DiMarco et al., 2023), and synteny”

The discussion seems to be very short and should be elaborated more. More explanation about the shortcomings of the existing methods and techniques can be added. A paragraph on how these shortcomings can be improved can be added.

Experimental design

no comment

Validity of the findings

no comment

·

Basic reporting

This article deals with an interesting topic related to corynebacterium pseudotuberculosis and comparative genomics. It has unambiguous professional English. Also, it has good literature references to provide enough context and a professional article structure regarding figures and tables. It also shares its raw data as an "SRA" ID from NCBI.
However, they need to share all scripting and bioinformatics pipelines as a GitHub repository or something similar to replicate their results.

Experimental design

This article fits well with a well-defined research question and rigorous investigation performed at a high technical level. However, its methods section is not described with enough detail and information to replicate. Practically, it lacks parameters and precise bioinformatic pipelines.

Validity of the findings

This article has robust and statistically sound bioinformatic analysis. Also, its conclusions are well stated and liked with the original research question.

Additional comments

Detailed comments and suggestions are in the attached file. Please revise them

---

## Round 0.2 · accepted · Accept

The authors have properly addressed the reviewers' concerns.

Reviewer 2 ·

Basic reporting

All my concerns have been addressed.

Experimental design

no comment

Validity of the findings

no comment

Additional comments

All my concerns have been addressed.

·

Basic reporting

Authors have addressed well all the comments and suggestions from the reviewers.

Experimental design

Authors have addressed well all the comments and suggestions from the reviewers.

Validity of the findings

Authors have addressed well all the comments and suggestions from the reviewers.

Additional comments

Now this manuscript is suitable for their publication at PeerJ. Please make a double check for all typo mistakes, that's all.